# Influence of Alpine Skiing on Health-Related Quality of Life and Physical Self-Concept in Physically Active Adults over 55 Years of Age

**DOI:** 10.3390/sports10100153

**Published:** 2022-10-13

**Authors:** Javier Conde-Pipó, Ignacio Valenzuela-Barranco, Alejandro López-Moro, Blanca Román-Alconchel, Miguel Mariscal-Arcas, Félix Zurita-Ortega

**Affiliations:** 1Department of Didactics of Musical, Plastic and Corporal, Faculty of Education Sciences, University of Granada, 18071 Granada, Spain; 2Cetursa Sierra Nevada, Sierra Nevada-Monachil, 18196 Granada, Spain; 3Department Nutrition and Food Science, School of Pharmacy, University of Granada, Campus of Cartuja s/n, 18071 Granada, Spain

**Keywords:** psychosocial factors, physical activity, physical functioning, older adults, healthy ageing

## Abstract

*Background:* Older adults have the highest rates of a sedentary lifestyle. Alpine skiing could be considered a suitable activity to encourage continued sports practice and improve the health of this population in winter. The aim of this study was to analyse the relationship between the practice of alpine skiing and health-related quality of life (HRQoL), physical self-concept (PSC), and sport motivation. *Methods:* The study design was cross-sectional and descriptive, involving 280 Spanish adults aged over 55 years and physically active (75.35% skiers). To assess physical activity, PSC, HRQoL, and sport motivation, we used the Rapid Assessment of Physical Activity Questionnaire (RAPA-Q), the Physical Self-Perception Profile (PSPP 30), the Health-Related Quality of Life (SF-36), and the Sport Motivation Scale (SMS) questionnaires, respectively. *Results*: In the skier’s group, higher values were found for PSC (*p* < 0.001; d = 0.64), the physical component of HRQoL (*p* < 0.001, d = 0.48), physical function (*p* < 0.001, d = 61), and intrinsic motivation (*p* < 0.001; d = 0.85). The practice of alpine skiing was associated with higher levels of the physical health component (ORadj = 2.13, 95% CI 1.18–3.95, *p* = 0.013), PSC (ORadj = 2.92, 95% CI 1.58–5.52, *p* < 0.001), and intrinsic motivation (ORadj = 2.24, 95% CI 1.22–4.23, *p* = 0.010). *Conclusions:* The practice of alpine skiing is positively associated with higher values of HRQoL, PSC, and intrinsic motivation, and based on the above, it seems that alpine skiing can contribute to healthy ageing and improve the quality of life of older adults.

## 1. Introduction

Socio-economic progress and advances in medicine and public health are facilitating increases in life expectancy worldwide [1]. To minimise the risk of disease and disability, enjoy this increased longevity with a good quality of life, and at the same time, reduce the costs of maintaining public health and care services, it is necessary to encourage the older population to adopt healthy lifestyles based on proper nutrition and participation in regular physical activity (PA) [2,3]. For older adults, a minimum of 150–300 min of moderate–vigorous physical activity per week, including aerobic workout, strength, and balance, is recommended [4]. Despite this, the older population group has the highest rates of sedentary lifestyle [5]. During the winter months, the problem is exacerbated, as PA is reduced by up to 40% due to harsher environmental conditions [6]. It is therefore advisable to identify and promote those activities that are most attractive and effective in terms of continued practice, health, and quality of life for this population group.

Health-related quality of life (HRQoL) is an indicator that encompasses both physical and mental functioning [7] and is widely used to monitor population health [8]. There is a growing body of empirical evidence of a positive association between HRQoL and PA in general, although studies focusing on specific sports modalities are scarce, and cross-sectional studies are still mostly inconclusive [8,9,10,11,12,13,14,15]. Understanding how these variables are related globally is important, as it could facilitate the development of more effective interventions.

Physical self-concept (PSC) and sport motivation are two psychosocial factors of special interest for this study because they mediate adherence to sport and because of their relationship with quality of life [12,16], which is why both concepts are introduced below.

PSC, a subdomain of self-concept [17], is the set of ideas that we believe define us physically [18] and is composed of four dimensions: physical condition (PC), sport competence (SC), physical attractiveness (AT), and physical strength (ST). PSC has a significant influence on a person’s psychological well-being and is associated with healthy behaviours [12,19]. There is a positive association between PSC and PA, in general, that has been widely documented, although, in the case of older adults, studies are still scarce [20].

Motivation is understood as the force that energizes and directs behaviour [16]. Its most self-determined form, intrinsic motivation (IM), is of utmost importance for sports practice, as it acts as a mediator, determining the initiation, maintenance, and abandonment of sport [16]. Intrinsic motivations, such as enjoyment or considering sport as a significant part of one’s lifestyle, support adherence to PA, and are considered a predictor of the positive psychological, social and affective consequences of PA in adults [21,22,23].

The so-called lifetime sports are those that can be practised at any age and in which intrinsic motivations such as fun or health predominate, being the most practised by older adults [24]. Among the lifetime sports that can help increase the PA rate during winter is alpine skiing (AS), as it is one of the most popular outdoor sports activities at this time of the year [25]. Despite the complexity of learning it and not being exempt from certain risks [26], it is a sport that allows a very early initiation, which subsequently facilitates high adherence to the practice [27], with a very low dropout rate [28], and which complies with all sports practice recommendations for the older adults [29,30], even if they have chronic diseases [31].

Numerous findings have been found in recent years on the physical health benefits of AS. It is considered an intervallic strength-endurance sport performed at medium-high intensity [32], which mainly generates improvements in cardiorespiratory fitness and in the strength of the main locomotor muscles [31,32,33,34,35,36,37]. Regarding mental health, studies are still scarce, but it seems that AS could contribute to maintaining and improving mental health in adults. Improvements have been observed in some psychosocial factors, such as mood, self-concept, levels of depression, and levels of life satisfaction [32,38,39].

According to the above, AS could be considered a suitable activity to encourage the practice of sports in winter and improve the physical and mental health of older adults. Given that there are no studies to date that have analysed and compared HRQoL and the psychosocial factors described above in different types of sports, this research aims to fill this gap and help determine which sporting activity can contribute most significantly to the improvement of these variables.

In this sense, the main objective of this study was to analyse the relationship between the practice of AS and HRQoL, PSC, and motivation towards sports practice in a physically active Spanish population over 55 years of age. The secondary objective was to verify whether there is any association between the study variables and the practice of other sports specialities.

## 2. Materials and Methods

### 2.1. Design and Sample

The study design was cross-sectional, descriptive, and comparative. According to the data from the Spanish National Institute of Statistics [40] (https://www.ine.es/, 14 April 2022) on the Spanish population aged 55 years and over, it was estimated that a sample size of 273 participants would be sufficient under the conditions of α=0.05 and a two-tailed confidence interval = 90%. The sample was initially composed of 351 subjects, all of them from different regions of Spain, with the initial inclusion criteria being a minimum of 55 years of age and leading a physically active life [4]. Seventy of them were discarded for not meeting these criteria or not completing the questionnaires correctly, leaving a final sample of 280 subjects.

All participants were randomly recruited over three months, invited to participate voluntarily, and provided their written informed consent. All were informed of the objectives of this research and data protection was assured. The surveys were completed anonymously. The research complies with the principles of the Declaration of Helsinki and has the approval of the Research Ethics Committee of the University of Granada, with code 1230/CEIH/2020.

### 2.2. Instruments

#### 2.2.1. Physical Activity and Alpine Skiing

PA was assessed using the Spanish version of the Rapid Assessment of Physical Activity Questionnaire (RAPA-Q) [41], a validated seven-item questionnaire that is easy to use, has proven reliability and sensitivity, and is specifically designed for use with adults over 50 years of age. This questionnaire can be answered affirmatively (yes) or negatively (no) and allows easy identification of the level of PA according to whether the WHO recommendations of minimum practice for cardiovascular health benefits are achieved [4]. Thus, the level is classified as low when light to moderate activities are practised but not every week, moderate when these activities are practised less than 150 min per week or 75 min in the case of vigorous activities, or high when these time thresholds are exceeded [42]. This classification made it possible to group those subjects with a high level of PA as active, and the rest as non-active.

Two more items were added to this questionnaire on the number of sports practised and which were these, later grouping them into outdoor sports (hiking, running, cycling, etc.), maintenance sports (swimming, weights, Pilates, yoga, dance, etc.) or opponent sports (tennis, paddle tennis, team sports). In addition, they were asked whether they practised alpine skiing and, if so, the number of days per season, allowing subjects to be classified as skiers if they skied more than five days per season (SG) or non-skiers (NSG) [43].

#### 2.2.2. Physical Self-Concept

The adaptation of the PSPP 30 (Physical Self-Perception Profile) to the Spanish population carried out by Goñi [44] (original by Fox and Corbin [45]) was used. It is made up of 30 items that are valued on a five-point Likert scale (“Totally disagree” = 1, “Totally agree” = 5). It is divided into five dimensions: physical condition (PC, α = 0.84), sports competence (SC, α = 0.88), physical attractiveness (AT, α = 0.88), physical strength (ST, α = 0.83), and general physical self-concept (PSC, α = 0.88). Subsequently, the PSC variable was categorized into two levels, “high” and “low”, according to a score equal to or less than the 50th percentile of the corresponding gender, or greater than it. In our study, all the dimensions show good internal consistency, with Cronbach’s α coefficients ranging between 0.72 and 0.93.

#### 2.2.3. Heath-Related Quality of Life

The Spanish version of the SF-36 [46] questionnaire was used, which has proven validity, high reliability, and wide use in the older adult population for measuring their subjective health and capabilities or limitations to perform daily tasks [42,46,47]. It is composed of eight domains: physical function (PF, 10 items, α = 0.93), physical role (PR, 4 items, α = 0.95), bodily pain (BP, 2 items, α = 0.87), general health (GH, 5 items, α = 0.79), vitality (VI, 4 items, α = 0.85), social function (SF, 2 items, α = 0.72), emotional role (ER, 3 items, α = 0.91) and mental health (MH, 5 items, α = 0.85). Each domain is scored on a scale from 0 to 100, with the highest score corresponding to the best health status. It also provides two summary components, the physical component (COMP-P) and the mental component (COMP-M), which are calculated with weights specific to the Spanish population [48,49]. To establish the cut-off points that made it possible to classify both summary components into two levels, the medians of the tables of Spanish population norms [49,50] according to age groups and gender, were used.

In our study, the values of Cronbach’s α coefficients of the SF-36 ranged between 0.75 and 0.85, suggesting that the questionnaire had an acceptable internal consistency.

#### 2.2.4. Sport Motivation

The original SMS questionnaire by Pelletier [51], adapted to Spanish by Balaguer [52], was used. It is made up of three main dimensions: intrinsic motivation (IM, α = 0.79), extrinsic motivation (EM, α = 0.69), and amotivation (DM, α = 0.75). It consists of 28 items rated on a Likert scale, ranging from 1 (“It has nothing to do with me”) to 7 (“It totally fits me”). It is an instrument with adequate psychometric properties and is reliable and valid for the study of different types of motivational regulations in sports [16]. The α values obtained in our study show good reliability, ranging between 0.72 and 0.96. Subsequently, the IM variable was categorised into two levels, “high” or “low”, depending on whether the score was equal to or below the median of the corresponding gender, or above it.

#### 2.2.5. Sociodemographic Determinants

The researchers created an ad-hoc questionnaire for the collection of sociodemographic data and the medical history of the participants. They were asked about their gender, weight, and height (from which their BMI was estimated), and whether they had high blood pressure, cholesterol, or diabetes in the past 12 months.

### 2.3. Statistical Analysis

Statistical analysis was performed with the R statistical computing software (R Core Team, Vienna, Austria). The normality of the variables was analysed using the Kolmogorov–Smirnov test with Lillieforts correction and homoscedasticity with the Levene test. The frequencies, means, and standard deviations were used for basic descriptions. For comparisons between groups of continuous variables, the non-parametric Mann–Whitney U test was used for independent variables, and Cohen’s-d index was used to calculate the effect size. For comparisons between groups of categorical variables, the Pearson Chi-square test was used. For bivariate correlations, Spearman’s rho correlation coefficient was used. Logistic regression models were applied to assess the association between the prevalence of high levels of COMP-P and COMP-M, PSC, and IM with alpine skiing, outdoor sports, competitive sports, and maintenance sports. The results are presented as the crude (ORcr) and adjusted (ORadj) odds ratios with 95% confidence intervals.

The internal reliability of the instruments was assessed using Cronbach’s Alpha coefficient. All reported *p*-values are based on the two-tailed test, and the level of statistical significance for all tests was set at 95%.

## 3. Results

Table 1 shows the initial characteristics of the sample after being classified into skiers (*n* = 211; 61.23 years (SD: 5.4); males 84.83%; females 15.17%) and non-skiers (*n* = 69; 62.62 years (SD: 6.53); males 52.18%; females 47.82%). No statistically significant differences were found between both groups regarding health history (*p* > 0.05), although the BMI was significantly lower in skiers (25.86 Kg/m^2^ (SD: 3.27) vs. 27.06 Kg/m^2^ (SD: 7.33); *p* = 0.039). In terms of sports habits, the group of skiers was the one with the greatest variety of different sports practices (49.75% vs. 7.24%, *p* < 0.001), with outdoor sports and maintenance sports the most practised in both groups.

Table 2 shows the correlations between the dimensions of the PSC-Q, SF36 and IM differentiated by group (SG vs. NSG). In the SG, there were a higher number of HRQoL dimensions that correlated significantly (*p* ≤ 0.001) with PSC compared to the NSG (PF, r = 0.27; GH, r = 0.34; VI, r = 0.30; MH, r = 0.17 vs. GH, r = 0.32; VI, r = 0.31). Regarding the dimensions of PSC, among the skiers, the correlations shown by strength with respect to GH (r = 0.38) and by PC with respect to PF (r = 0.35) stand out, while in the NSG, PC with respect to GH (r = 0.50), VI (r = 0.45), and MH (r = 0.36) stand out.

Table 3 shows the mean values of the variables studied. All the means were higher in the EG group, with the differences being statistically significant except for the variables AT, BP, ER, COMP-M, and EM. The largest effect sizes were obtained in the variables PSC (d = 0.64; CI 0.34–0.93; *p* < 0.001), PF (d = 61; CI 0.31–0.90; *p* < 0.001), and IM (d = 0.85; CI 0.56–1.15, *p* < 0.001).

Table 4 shows the crude and adjusted results of the logistic regression models that evaluate the association between having a high level of COMP-P, COMP-M, PSC, and IM with the type of sport practiced. Alpine skiing was the only sport that was significantly associated with a higher level of COMP-P (ORadj = 2.13, 95% CI 1.18–3.95, *p* = 0.01), a higher PSC (ORadj = 2.92, 95% CI 1.58–5.52, *p* = 0.001), and higher IM (ORadj = 2.24, 95% CI 1.22–4.23, *p* = 0.010).

## 4. Discussion

The findings published here regarding HRQoL are analogous to those published in previous studies. They conclude that sports practice does not necessarily have an impact on all dimensions of HRQoL, with significant differences in physical function, vitality, and general health, as long as it is performed at high volumes or at vigorous intensities [8,13,15,53,54,55]. Therefore, there could be a threshold below which PA would not provide any benefit to HRQoL, making a difference in terms of health between the effectiveness of practising one sport or another, which must be taken into account when prescribing PA for the older adult population [50,53,54].

There are still few studies that have investigated HRQoL in the specific context of a sporting modality, but there is some evidence that low-intensity sports have little or no impact on HRQoL. This is evidenced by studies on yoga, tai chi, and Pilates [56,57,58]. However, in other studies carried out on aquatic sports at a higher intensity, more global improvements in HRQoL have been obtained [59,60].

This threshold theory could explain the differences found in our study between skiers and non-skiers. Skiing is a very complete sport practised over a whole day, in which acute cardiac responses that are beneficial for cardiovascular health [38,61,62,63] alternate with the necessary breaks to go up again taking a lift. The entire locomotor musculature is also subjected to intermittent efforts of medium and high intensities that favour the capacity to exert effort and perform daily tasks [37,64]. It also leads to an improvement in bone density [65,66], reduction in body fat, and insulin resistance [35]—aspects related to general health [32]. The summary of all these could justify the differences found in the PF, PR, and GH dimensions of HRQoL. The higher levels in the VI and SF dimensions could be due to the fact that alpine skiing is an activity that generates positive alterations in mental and social health [39] and is associated with life satisfaction and social well-being [32].

### 4.1. Physical Self-Concept

Previous research has already shown a positive association between PSC and continuous PA practice in general, the intensity at which it is performed [66,67], and the number of sports practised [20,68]. However, little is known about the impact that different sports modalities may have on PSC and its dimensions, and most research has focused on adolescents. Dolenc [69] compared individual sports with team sports, concluding that the former show higher values in the PC and SC dimensions. Esnaola [70], in another comparative study of various sports, found higher levels of PSC in rhythmic gymnastics and swimming, also with greater differences in PC and SC. These same dimensions are the ones that have obtained the best scores in other studies carried out on rhythmic gymnastics, aerobics, judo, and football [18,67,71,72,73].

In the present study, skiers showed a higher PSC, especially in the ST and SC dimensions, with a positive association between high levels of PSC and alpine skiing that was not found in other types of sports. These findings corroborate what has been established in previous studies in this population group, with this relationship explained by the improvement in strength levels and aerobic capacity [39,63,74,75]. There have been other studies with similar results. Amesberguer et al. [76], after intervening with an alpine skiing program in subjects with knee injuries, found that the changes in physical self-concept were positive but marginal. However, in subjects with spinal cord injuries, there was an evident improvement in PSC in the dimensions of AT, SC, and PC [77,78,79].

All these findings suggest that PSC could depend on the technical complexity and physical demands of each sport, characteristics which, as mentioned above, are very present in alpine skiing [27,64], and could justify the results obtained in the present study.

### 4.2. Intrinsic Motivation

Our results indicate that there is a positive association between alpine skiing and high IM values. These results are consistent with what has been established in the literature. On the one hand, for this age group, IM is predominant and favours sports practice [80]. On the other hand, adherence to alpine skiing seems to depend mainly on this type of motivation, such as the search for sensations, personal challenge, maintaining contact with friends, and enjoying nature [80,81,82]. Therefore, alpine skiing could contribute to frequent sports practice, and thus, to healthier ageing [32].

### 4.3. Limitations and Future Perspectives

To date, this is the first study to consider alpine skiing as a tool to promote the practice of physical activity during the winter months and to analyse its relationship with health parameters in the adult population over 55 years of age. Its main limitation is its descriptive and cross-sectional design, which does not allow for establishing causal relationships among the variables, so longitudinal studies and randomized control trials will be necessary in the future. On the other hand, the fact that alpine skiing may imply belonging to a higher socio-economic status could lead to a bias in the extrapolation of the results. In addition, the absence of objective measures for the assessment of PA makes it necessary to interpret these results with caution.

In consideration of the findings and limitations of this study, the following suggestions are made for future studies. This research should be replicated considering gender differentiation, socio-economic stratification, and other health parameters, such as nutritional ones, since the practice of alpine skiing is part of a lifestyle, and it is necessary to isolate other behaviours and activities that could be interacting with HRQoL or PSC.

## 5. Conclusions

Based on the interpretation of these data, we can conclude that, in Spain, the older adult alpine skiing population presents better parameters of both health-related quality of life and physical self-concept compared to non-skiers. Among these parameters, physical function and general health stand out with respect to quality of life, and strength and sporting ability stand out with respect to physical self-concept. Compared to other types of sports commonly practised by this segment of the population, alpine skiing is associated with higher levels of physical health, physical self-concept, and intrinsic motivation towards sport.

Accordingly, and in line with what has also been established in the scientific literature on the benefits of alpine skiing for cardiovascular health and strength maintenance, it seems plausible that this sport can contribute to healthy ageing and improve the quality of life of older adults.

### Practical Applications

From an applicative perspective, these findings support the convenience of encouraging the practice of alpine skiing among this population group through institutional programs for the promotion of sport and public health, and it is an appropriate strategy to achieve both greater adherence to sport during the months in which it declines and better health parameters.

## Figures and Tables

**Table 1 sports-10-00153-t001:** Characteristics of the study sample.

*n* = 280	Skiers	Non-Skiers	*p*
Distribution, n, %	211, 75.35	69, 24.65	0.001
Sex, n, %			
Men	179, 84.83	36, 52.18	0.001
Women	32, 15.17	33, 47.82	
Age, years, SD	61.23 (5.41)	62.62 (6.53)	0.137
Height, cm, SD	172 (7.57)	166 (10.93)	0.001
Weight, kg, SD	77.36 (11.02)	74.96 (13.58)	0.207
Body mass index, Kg/m^2^, SD	25.86 (3.27)	27.06 (7.33)	0.039
Hypertension *, n, %	35 (21.60)	10 (14.49)	0.585
Diabetes *, n, %	8 (4.94)	5 (7.24)	0.428
Cholesterol *, n, %	31 (19.14)	17 (17.69)	0.111
Number of sports practised, n, %			
1	12 (5.69)	39 (56.52)	0.001
2	94 (44.55)	16 (23.18)	
3 or more	105 (49.75)	5 (7.24)	
Type of sports practised, n, %			
Outdoor *	164 (77.73)	38 (55.07)	0.037
Opponent *	32 (15.17)	14 (20.28)	0.173
Maintenance *	105 (49.76)	28 (40.57)	0.767

* Affirmative answers.

**Table 2 sports-10-00153-t002:** Correlations between the PSC-Q, SF36 and intrinsic motivation questionnaires differentiated by skiers and non-skiers.

Non-Skiers														
Variable	1	2	3	4	5	6	7	8	9	10	11	12	13	14
Skiers														
1		**0.44**	**0.64**	**0.44**	**0.64**	0.12	0.00	0.03	**0.32**	**0.31**	0.21	−0.08	0.19	**0.28**
		(0.21, 0.63)	(0.21, 0.63)	(0.46, 0.77)	(0.21, 0.62)	(0.46, 0.64)	(−0.26, 0.25)	(−0.23, 0.28)	(0.08, 0.53)	(0.06, 0.44)	(−0.05, 0.44)	(−0.32, 0.18)	(−0.06, 0.43)	(0.03, 0.50)
2	**0.46**		**0.69**	**0.73**	**0.70**	0.22	0.15	0.21	**0.50**	**0.45**	**0.38**	−0.08	**0.36**	**0.59**
	(0.34, 0.56)		(0.53, 0.80)	(0.55, 0.83)	(0.55, 0.81)	(−0.03, 0.459	(−0.11, 0.39)	(−0.05, 0.44)	(0.29, 0.67)	(0.22, 0.63)	(0.14, 0.58)	(−0.32, 0.18)	(0.12, 0.56)	(0.40, 0.74)
3	**0.54**	**0.59**		**0.56**	**0.65**	0.12	0.15	0.14	**0.44**	**036**	**0.34**	−0.06	0.22	**0.19**
	(0.45, 0.64)	(0.50, 0.67)		(0.36, 0.71)	(0.48, 0.78)	(−0.14, 0.36)	(−0.11, 0.39)	(−0.12, 0.38)	(0.21, 0.62)	(0.12, 0.56)	(0.09, 0.54)	(−0.31, 0.19)	(−0.04, 0.45)	(0.24, 0.64)
4	**0.54**	**0.66**	**0.46**		**0.64**	0.22	0.19	**0.26**	**0.40**	**0.41**	**0.36**	0.01	**0.28**	**0.54**
	(0.44, 0.63)	(0.57, 0.73)	(0.35, 0.56)		(0.47, 0.77)	(−0.04, 0.45)	(−0.07, 0.42)	(0.00, 0.48)	(0.17, 0.60)	(0.17, 0.60)	(0.12, 0.56)	(−0.25, 0.26)	(0.02, 0.50)	(0.33, 0.70)
5	**0.46**	**0.59**	**0.53**	**0.49**		0.07	0.13	0.6	**0.41**	**0.37**	**0.39**	**−0.03**	**0.32**	**0.30**
	(0.35, 0.56)	(0.49, 0.67)	(0.42, 0.62)	(0.38, 0.59)		(−0.19, 0.32)	(−0.13, 0.37)	(−0.10, 0.40)	(0.18, 0.60)	(0.13, 0.57)	(0.15, 0.58)	(−0.28, 0.23)	(0.07, 0.53)	(0.05, 0.52)
6	**0.27**	**035**	**0.27**	**0.30**	**0.30**		**0.49**	**0.45**	**0.47**	**0.43**	**0.41**	**0.41**	**0.35**	**0.28**
	(0.14, 0.39)	(0.23, 0.46)	(0.14, 0.39)	(0.17, 0.42)	(0.18, 0.42)		(0.27, 0.66)	(0.22, 0.63)	(0.25, 0.65)	(0.20, 0.62)	(0.17, 0.60)	(0.17, 0.60)	(0.11, 0.56)	(0.02, 0.50)
7	0.09	0.12	**0.19**	0.09	0.05	**0.40**		**0.45**	**0.45**	**0.52**	**0.58**	**0.58**	**0.45**	0.04
	(−0.05, 0.22)	(−0.12, 0.25)	(0.05, 0.31)	(−0.05, 0.22)	(−0.09, 0.18)	(0.28, 0.51)		(0.22, 0.63)	(0.23, 0.63)	(0.30, 0.68)	(0.38, 0.73)	(0.39, 0.73)	(0.22, 0.63)	(−0.21, 0.29)
8	0.02	0.06	0.10	−0.01	0.06	**0.40**	**0.34**		**0.47**	**0.47**	**0.44**	0.21	0.44	0.07
	(−0.11, 0.16)	(−0.08, 0.19)	(−0.04, 0.23)	(−0.14, −0.01)	(−0.08, 0.19)	(0.28, 0.51)	(0.21, 0.45)		(0.24, 0.73)	(0.24, 0.73)	(0.21, 0.63)	(−0.04, 0.44)	(0.21, 0.63)	(−0.19, 0.32)
9	**0.34**	**0.30**	**0.36**	**0.21**	**0.38**	**0.34**	**0.31**	**0.30**		**0.67**	**0.47**	0.22	**0.45**	**0.43**
	(0.22, 0.46)	(0.17, 0.41)	(0.24, 0.48)	(0.08, 0.34)	(0.26, 0.49)	(0.22, 0.46)	(0.18, 0.42)	(0.17, 0.42)		(0.50, 0.79)	(0.25, 0.65)	(−0.03, 0.45)	(0.22, 0.63)	(0.19, 0.61)
10	**0.30**	**0.30**	**0.40**	**0.30**	**0.29**	**0.34**	**0.35**	**0.36**	**0.40**		**0.43**	**0.38**	**0.61**	**0.26**
	(0.03, 0.29)	(0.17, 0.42)	(0.28, 0.50)	(0.18, 0.42)	(0.17, 0.41)	(0.21, 0.45)	(0.23, 0.47)	(0.24, 0.47)	(0.28, 0.51)		(0.20, 0.62)	(0.14, 0.58)	(0.42, 0.74)	(0.00, 0.48)
11	**0.16**	0.05	**0.17**	0.11	**0.16**	**0.28**	**0.34**	**0.24**	**0.22**	**0.42**		**0.33**	**0.51**	0.12
	(0.03, 0.29)	(−0.09, 0.18)	(0.03, 0.30)	(−0.03, 0.24)	(0.03, 0.29)	(0.15, 0.40)	(0.22, 0.45)	(0.11, 0.36)	(0.09, 0.34)	(0.30, 0.53)		(0.09, 0.54)	(0.29, 0.67)	(−0.14, 0.36)
12	0.06	−0.03	0.09	−0.03	0.05	013	**0.38**	0.10	**0.20**	**0.29**	**0.36**		**0.43**	−0.16
	(−0.08, 0.19)	(−0.16, 0.11)	(−0.05, 0.22)	(−0.17, 0.10)	(−0.08, 0.19)	(0.00, 0.26)	(0.26, 0.49)	(−0.03, 0.24)	(0.07, 0.32)	(0.16, 0.41)	(0.24, 0.48)		(0.20, 0.61)	(−0.40, 010)
13	**0.17**	0.09	**0.21**	**0.13**	0.07	**0.16**	**0.22**	**0.15**	**0.30**	**0.55**	**0.40**	**0.39**		0.14
	(0.04, 0.30)	(−0.04, 0.23)	(0.07, 0.33)	(0.00, 0.26)	(−0.06, 0.21)	(0.03, 0.29)	(0.09, 0.34)	(0.02, 0.28)	(0.17, 0.42)	(0.45, 0.64)	(0.28, 0.50)	(0.27, 0.50)		(−0.12, 0.38)
14	**0.30**	**0.29**	**0.22**	**0.34**	**0.28**	0.10	−0.02	−0.02	0.26	0.19	0.04	−0.09	0.12	
	(0.17, 0.42)	(0.16, 0.41)	(0.09, 0.35)	(0.21, 0.45)	(0.05, 0.40)	(−0.03, 0.23)	(−0.16, 0.11)	(−0.16, 0.11)	(0.13, 0.39)	(0.24, 0.48)	(−0.09, 0.18)	(−0.022, 0.05)	(−0.01, 0.25)	

Note: 1: physical self-concept; 2: physical condition; 3: physical attractiveness; 4: sport competence; 5: strength; 6: physical function; 7: physical role; 8: bodily pain; 9: general health; 10: vitality; 11: social function; 12: emotional role; 13: mental health; 14: intrinsic motivation; significant values by Spearman’s rho correlation test are indicated in bold.

**Table 3 sports-10-00153-t003:** Results of the PSC, HRQoL, and SMS questionnaires and their association with alpine skiing.

Questionnaire	Variables	Skiers	Non-Skiers	*p*	Effect Size	
		M (SD)	M (SD)		d	CI
PSC-Q	PSC	15.98 (2.66)	14.13 (3.63)	0.001	0.64	(0.34, 0.93)
	ST	14.67 (2.81)	13.07 (3.76)	0.001	0.53	(0.24, 0.82)
	SC	15.69 (3.28)	13.93 (4.31)	0.003	0.50	(0.21, 0.79)
	AT	23.00 (4.49)	21.97 (5.25)	0.253	0.22	(0.07, 0.51)
	PC	17.11 (3.77)	15.63 (3.97)	0.022	0.39	(0.10, 0.67)
SF-36	Comp-P	50.59 (7.08)	47.17 (7.50)	0.001	0.48	(0.19, 0.77)
	Comp-M	51.10 (9.39)	48.95 (10.27)	0.161	0.22	(0.06, 0.51)
	PF	92.70 (10.70)	85.83 (13.31)	0.001	0.61	(0.31, 0.90)
	RF	66.23 (37.40)	51.67 (44.36)	0.041	0.37	(0.08, 0.66)
	BP	74.60 (19.92)	68.90 (20.57)	0.080	0.28	(0.00, 0.57)
	GH	77.23 (14.49)	68.60 (16.62)	0.001	0.58	(0.28, 0.87)
	VI	78.79 (13.20)	73.00 (16.58)	0.044	0.52	(0.10, 0.94)
	SF	90.05 (18.38)	84.38 (19.07)	0.016	0.41	(0.12, 0.70)
	ER	71.72 (38.29)	65.56 (44.24)	0.461	0.16	(0.13, 0.44)
	MH	83.56 (13.27)	77.40 (18.08)	0.018	0.43	(0.14, 0.71)
SMS	IM	5.68 (1.02)	4.68 (1.61)	0.001	0.85	(0.56, 1.15)
	EM	3.85 (1.22)	3.96 (1.23)	0.401	0.09	(0.20, 0.37)
	NM	2.17 (1.16)	2.70 (1.14)	0.001	0.46	(0.17, 0.75)

Note: ACF: physical self-concept; ST: strength; SC: sport competence; AT: physical attractiveness; PC: physical condition; Comp-P: physical component; Comp-M: mental component; PF: physical function; RF: physical role; BP: bodily paint; GH: general health; SF: social function; ER: emotional role; MH: mental health; IM: intrinsic motivation; EM: extrinsic motivation; NM: non-motivation. M: mean; SD: standard deviation; CI: confidence interval.

**Table 4 sports-10-00153-t004:** Association between high level of Comp-P, Comp-M, PSC, and IM with different types of sport.

	Physical Component (Comp-P)						Mental Component (Comp-M)					
Factors	OR_cr_	CI (95%)	*p*	OR_adj_	CI (95%)	*p*	OR_cr_	CI (95%)	*p*	OR_adj_	CI (95%)	*p*
Alpine skiing	2.05	(1.14–3.75)	0.017	2.13	(1.18–3.95)	0.013	1.52	(0.85–2.74	0.154	1.51	(0.84–2.73)	0.172
Outdoors	0.84	(0.48–1.45)	0.531	0.84	(0.45–1.55)	0.583	1.11	(0.64–1.93)	0.702	0.90	(0.49–1.65)	0.742
Maintenance	1.39	(0.86, 2.24)	0.180	1.35	(0.82–2.20)	0.232	0.87	(0.54–1.41	0.584	0.85	(0.52–1.39)	0.537
Opponent	1.21	(0.64, 2.31)	0.687	1.22	(0.61–2.47)	0.575	0.66	(0.34–1.25)	0.207	0.66	(0.32–1.31)	0.238
	Physical Self-Concept (PSC)						Intrinsic Motivation (IM)					
Factors	OR_cr_	CI (95%)	*p*	OR_adj_	CI (95%)	*p*	OR_cr_	CI (95%)	*p*	OR_adj_	CI (95%)	*p*
Alpine skiing	2.95	(1.63–5.49)	0.015	2.92	(1.58–5.52)	0.001	1.82	(0.77–4.37)	0.172	2.24	(1.22–4.23)	0.010
Outdoors	1.37	(0.79–2.39)	0.258	1.64	(0.87–3.10)	0.124	1.17	(0.47–2.88)	0.734	1.01	(0.55–1.87)	0.967
Maintenance	1.96	(1.21–3.20)	0.794	2.08	(1.26–3.48)	0.004	1.09	(0.50–2.38)	0.826	1.28	(0.78–2.09)	0.328
Opponent	1.23	(0.65–2.37)	0.548	1.72	(0.834–3.68)	0.147	1.05	(0.39–2.81)	0.924	1.12	(0.55–2.26)	0.745

Note: Comp-P: physical component of HRQoL; Comp-M: mental component of HRQoL; PSC: physical self-concept; IM: intrinsic motivation; OR_cr_: OR crude; OR_adj_: OR adjusted; CI: confidence interval.

## Data Availability

There are restrictions on the availability of the data for this trial due to the signed consent agreements around data sharing, which only allow access to external researchers for studies following the project’s purposes. Requestors wishing to access the trial data used in this study can make a request to mariscal@ugr.es+.

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
