# Peer review of "Influence of Alpine Skiing on Health-Related Quality of Life and Physical Self-Concept in Physically Active Adults over 55 Years of Age"

_sports, 2022, doi:10.3390/sports10100153_

Round 1
Reviewer 1 Report
Older adults have the highest rates of sedentary lifestyle. Alpine skiing could be consid- 14 ered a suitable activity to encourage continued sports practice and improve the health of this pop- 15 ulation in winter. The aim of this study was to analyse the relationship between the practice of alpine 16 skiing and health-related quality of life (HRQoL), physical self-concept (PSC) and sport motivation 17 in Spanish adults aged over 55 years and physically active.
1) suggests dividing the abstract into (backgroud, methods, results, conclusion)
2) In the introduction, you can do a more detailed literature review, it is worth mentioning here the so called "lifetime sports" (cross-country skiing, cycling, Nordic walking)
3) Were there any inclusion and exclusion criteria in your study? It is worth mentioning this
4) The statistical analysis is carried out appropriately, although in the presentation of the results I suggest the use of bold font for statistically significant values especially in Table 2. Also the presentation of Table 4 is quite illegible especially in the confidence intervals.
5. add practical application of your research in the form of application conclusions
Author Response
Older adults have the highest rates of sedentary lifestyle. Alpine skiing could be considered a suitable activity to encourage continued sports practice and improve the health of this population in winter. The aim of this study was to analyse the relationship between the practice of alpine skiing and health-related quality of life (HRQoL), physical self-concept (PSC) and sport motivation 17 in Spanish adults aged over 55 years and physically active.
- suggests dividing the abstract into (backgroud, methods, results, conclusion)
Changes have been made in the abstract to adapt it to the proposed structure.
2) In the introduction, you can do a more detailed literature review, it is worth mentioning here the so called "lifetime sports" (cross-country skiing, cycling, Nordic walking)
A brief reference to this type of sports has been added, among which is alpine skiing (lines 66-69).
3) Were there any inclusion and exclusion criteria in your study? It is worth mentioning this
The initial inclusion criteria were being a minimum of 55 years of age and leading a physically active life, both mentioned in lines 100-101
4) The statistical analysis is carried out appropriately, although in the presentation of the results I suggest the use of bold font for statistically significant values especially in Table 2. Also the presentation of Table 4 is quite illegible especially in the confidence intervals.
Your suggestions have been included and the presentation has been improved.
- add practical application of your research in the form of application conclusions
Practical applications are included in lines 343-347

Reviewer 2 Report
Reviewer’s comments to author
The present study was designed to assess “the relationship between the practice of alpine skiing and the variables of health-related quality of life (HRQoL), physical self-concept (PSC) and sport motivation in Spanish adults aged over 55 years”. Researchers here followed a cross-sectional design and a quantitative approach to investigate the above research questions. The present study may represent an important addition to the existing literature as no other study has examined the aforementioned research hypothesis. Therefore, I believe it is publishable in its current form with a major revision. Below, I offer both general and specific comments regarding the current manuscript and, where appropriate, I have made suggestions to the authors regarding potential strategies for addressing these tasks.
General Comments:
· A innovative topic
· It covers the aims and the scope of this scientific journal
· It follows a cross-sectional design and a quantitative approach to investigate the research questions
· A very well written and easy to read text
Moreover,
· Please re-write all the reference number based on Sport based on the Sports journal (MDPI) format e.g., set reference number in brackets and not use reference number as superscript. As the following example: [1]
· Regarding References, please correct all the references based on the Sports journal (MDPI) format.
· In general, please follow the format style of the Sports journal (MDPI).
Specific comments:
Abstract
· Page 1, lines 19-20: Please delete the following phrase: “Physical activity was evaluated using 19 the Rapid Assessment of Physical Activity Questionnaire.”
· Page 1, lines 20-21: … “RAPA-Q, PSPP 30, SF-36 and SMS questionnaire …” - Please write here the full name of these instruments and set the shortcuts into brackets e.g. (RAPA-Q). You can also present here the reference number of each instrument.
· Page 1, lines 22-24: Please rewrite the following sentence in a proper way - “Stand out the higher values of PSC (p<0.001; d=0.64), physical component of HRQoL (p<0.001, d=0.48), physical function (p<0.001, d=61), and intrinsic motivation (p<0.001; d=0.85) found in skier´s group.”
· Page 1, lines 26-28: “The practising of alpine skiing is positively associated with higher values of HRQoL, PSC and intrinsic motivation and can contribute to a healthy ageing and improve the quality of life of older adults.” - Please rewrite the above sentence in a proper way e.g. “Based on the above …” or “To conclude …” “… it seems that that alpine skiing can contribute to a healthy ageing and improve the quality of life of older adults.”
1. Introduction
· Page 1, line 31: Please add numbering before the sub-titles e.g., “1. Introduction”
· Page 1, lines 36-37: …and physical activity (PA)..” – Please change the above phrase as” … “and participation in regular physical activity (PA) [2,3].
2. Material and methods.
· Page 2, line 91: Please add numbering before the sub-titles e.g., “2.1. Design and sample”
· Page 3, line 98: Please change the number 98 with the word “Ninety-eight”. You can then mention the number into a bracket e.g. “Ninety-eight (n = 98) …”
· Page 4, line 107: From the phrase “Instruments and variables.” please delete “… and variables.”. Please keep only “Instruments” and use also italics. Please add numbering before the sub-titles e.g., “2.2. Instruments”
· Page 4, line 108: Please add numbering before the sub-titles e.g., “2.2.1. Physical activity and alpine skiing”
· Page 4, line 110: Please delete comma after the word “Questionnaire,”
· Page 4, line 119: Please change “physical activity” with “PA”.
· Page 4, line 120: Please change the phrase “Two other questions…” (e.g. “Two more items …”)
· Page 4, line 126: Please add numbering before the sub-titles e.g., “2.2.2. Physical Self-Concept”
· Page 4, line 131: Please change 50th to 50th (superscript)
· Page 4, line 137: Please add numbering before the sub-titles e.g., “2.2.3. Heath related Quality of Life”
· Page 4, line 153: Please add numbering before the sub-titles e.g., “2.2.4. Sport motivation”
· Page 5, lines 166-167: Please correct the phrase “… whether they had or had had high blood pressure ….”
· Page 4, line 168: Please add numbering before the sub-titles e.g., “2.2.5. Statistical analysis”
3. Results
· Page 5, line 184: Please add numbering before the sub-titles e.g., “3. Results”
· Page 5, line 186: Please change table with Table. Also, please do not start with a number the sentence (75.35% ….). Try to change it.
· Page 5, lines 186-188: Please use these kind of brackets here […] e.g. [n=211; 61.23 years (SD:5.4); males 84.83%; 186 females 15.17%] and 24.65% non-skiers [n=69; 62.62 years (SD:6.53; males 52.18%; females 187 47.82%]. Also, please add some empty space between the words e.g. (SD: 5.4) or (SD: 6.53)
· Page 5, lines 189-190: Please change m2 to m2 (superscript). Also, please add some empty space between the words e.g. 25.86 Kg/m2 or 27.06 Kg/m2
· Page 6, lines 194-195: Please correct the alignment of the first column at Table 1.
· Page 6, line 208: Please change P with p (no capitals)
· Page 6, lines 209-210: Please add an empty space after Table 2 and the next paragraph
· Page 6, Table 3: Please change the acronym IC with CI. Please add an explanation at the bottom of Table 3. Also, add an explanation at the bottom of Table 3 for M and SD
· Page 7, Table 4: Please correct the alignment of Table 4 (e.g., use a landscape background) Please change the acronym IC with CI. Please add an explanation at the bottom of Table 4
Discussion
· Page 7, line 232: Please add numbering before the sub-titles e.g., “4. Discussion”
· Page 8, line 260: Change “… the summary of all this ...” with “... the summary of all these …”
· Page 8, line 286: Please change the phrase “… in our study” with “… in the present study.”
· Page 8, line 297: Please change “physical activity” with “PA”.
· Page 8, line 300: You could add after “… so longitudinal studies …” “… so longitudinal studies and randomized control trials …”
· Page 9, line 303: Change “physical activity” with “PA”.
Conclusions
· Page 9, line 310: Please add numbering before the sub-titles e.g., “5. Conclusions”
Author Contributions
· Page 9, line 334: Please use bold letters for the phrase “Author Contributions:”
References
· Pages 10-13, lines 357-541: Please follow the guidelines of Sports (MDPI) regarding the reference writing. Here you will probably need to do extensive changes.
Reviewer’s Decision: Accepted with major revision

Author Response
We appreciate your thorough review and comments, all proposed changes have been made.

Round 2
Reviewer 1 Report
Thank you for your answers, however:
1.The abstract has not been divided into the listed sections
2.If statistically significant values are indicated in bold , remove the * indication
3.Practical application is worth indicating as a new subsection
Author Response
Reviewer 1, round 2:
1.The abstract has not been divided into the listed sections
The following sections have been included: background, methods, results, and conclusions
2.If statistically significant values are indicated in bold, remove the * indication
All * have been deleted and an explanation add at the bottom.
3.Practical application is worth indicating as a new subsection
A new subsection has been added for practical application.

Reviewer 2 Report
Dear authors,
thank you very much for the improvements you have done in your manuscript. Now, I believe it has been sufficiently improved to warrant publication in Sports. I have only one more change to suggest:
Page 3, line 112: Please delete dot (.) after the sub-title 2.2. Instruments
Author Response
Reviewer 2, round 2:
Page 3, line 112: Please delete dot (.) after the sub-title 2.2. Instruments
The dot has been deleted.
